# Advanced Management Protocol of Transanal Irrigation in Order to Improve the Outcome of Pediatric Patients with Fecal Incontinence

**DOI:** 10.3390/children8121174

**Published:** 2021-12-11

**Authors:** Anna Maria Caruso, Mario Pietro Marcello Milazzo, Denisia Bommarito, Vincenza Girgenti, Glenda Amato, Giuseppe Paviglianiti, Alessandra Casuccio, Pieralba Catalano, Marcello Cimador, Maria Rita Di Pace

**Affiliations:** 1Pediatric Surgical Unit, Children’s Hospital ‘G. di Cristina’, ARNAS Civico-Di Cristina Benfratelli, 90100 Palermo, Italy; mariopietromarcello.milazzo@arnascivico.it (M.P.M.M.); denisia.bommarito@arnascivico.it (D.B.); vincenza.girgenti@arnascivico.it (V.G.); glenda.amato@arnascivico.it (G.A.); 2Pediatric Radiology Unit, Children’s Hospital ‘G. di Cristina’, ARNAS Civico-Di Cristina Benfratelli, 90100 Palermo, Italy; giuseppe.paviglianiti@arnascivico.it; 3Pediatric Surgical Unit, Department of Health Promotion, Mother and Child Care, Internal Medicine and Medical Specialities, University of Palermo, 90127 Palermo, Italy; alessandra.casuccio@unipa.it (A.C.); pieralba.catalano@libero.it (P.C.); marcello.cimador@unipa.it (M.C.); mariarita.dipace@unipa.it (M.R.D.P.)

**Keywords:** transanal irrigation, fecal incontinence, anorectal high-resolution manometry, children

## Abstract

Background: Transanal irrigation (TAI) is employed for children with fecal incontinence, but it can present several problems which require a study of their outcomes among different pathologies and without a tailored work up. The aim of our study was to evaluate the effectiveness of an advanced protocol in order to tailor TAI, prevent complications, and evaluate outcomes. Methods: We included 70 patients (14 anorectal malformation, 12 Hirschsprung’s disease, 24 neurological impairment, 20 functional incontinence) submitted to a comprehensive protocol with Peristeen^®^: fecal score, volumetric enema, rectal ultrasound, anorectal 3D manometry, and diary for testing and parameter adjustment. Results: Among the patients, 62.9% needed adaptations to the parameters, mainly volume of irrigated water and number of puffs of balloon. These adaptations were positively correlated with pre-treatment manometric and enema data. In each group, the improvement of score was statistically significant in all cases (*p* 0.000); the main factor influencing the efficacy was the rate of sphincter anomalies. The ARM group had slower improvement than other groups, whereas functional patients had the best response. Conclusions: Our results showed that TAI should not be standardized for all patients, because each one has different peculiarities; evaluation of patients before TAI with rectal ultrasound, enema, and manometry allowed us to tailor the treatment, highlighting different outcomes among various pathologies, thus improving the efficacy.

## 1. Introduction

Severe constipation and fecal incontinence are often observed in pediatric patients with either functional or organic bowel dysfunction; the management of this complex group of patients can be difficult and the different treatments proposed so far—either invasive or not—have shown variable efficacy [1,2,3,4]. Among them, transanal irrigation (TAI) can be proposed as a different conservative treatment also in children with bowel dysfunction unresponsive to more conventional therapies such as enema or laxatives. TAI involves a large volume water irrigation of the rectum and the left colon, passing a catheter with a balloon as an anchor through the anus [5,6].

In recent years, some data have been published [1,2,3,4,5,6] concerning the efficacy of transanal irrigation in the pediatric population but, although this treatment in adults has been well standardized, the practice in children still lacks empirical support; furthermore, complications or problems arising during the treatment often lead to interruption of the procedure for a brief period of time [7,8,9]. Despite the existence of valid but generic pediatric best practice [10], there are no series reporting on differentiated outcomes and comparisson among different pathologies. Another important issue concerns the management of patients before starting treatment with TAI. Since each pathology has peculiar physiopathological intestinal aspects, the aim of our study was to evaluate the effectiveness of an advanced management protocol in order to tailor and customize the TAI treatment for each patient, to prevent and manage the complications, and to evaluate the outcome considering the different pathologies.

## 2. Materials and Methods

### 2.1. Patients

A prospective analysis was conducted in our center from 2017 to 2021. Seventy patients up to 18 years of age, with fecal incontinence and/or bowel disfunction unresponsive to conventional treatments with laxative and enema, and with clinical indication to TAI, were included in the study. We included 14 patients operated on for anorectal malformation (ARM group), 12 patients for Hirschsprung’s disease (HD group), 24 patients with neurological impairment by cerebral palsy (NI group), and 20 patients with functional fecal incontinence (FFI group). We excluded patients who had undergone anorectal surgery or TAI in the past 6 months. We included only collaborating patients regardless of their age and family’s ability to objectively and clearly fill in the evaluation questionnaires and to carefully follow the instructions provided for the procedure. 

Among the 70 patients included, 24 had been treated with TAI in the past with little benefit and had stopped the treatment due to problems (mainly pain during the procedure, incontinence after the procedure, and incomplete evacuation) whereas 46 underwent TAI for the first time.

### 2.2. Protocol

All of our patients were submitted to a comprehensive protocol for TAI with the Peristeen Anal Irrigation System^®^ (Coloplast A/S Kokkedal, Humlebaek, Denmark) as follows:Clinical evaluation with Rintala continence score between 0 (very bad) and 20 (excellent). This score was calculated in baseline (T0) and during subsequent clinical check at 1, 3, and 6 months (T1, T3, and T6 respectively) [11].Ultrasound (US) evaluation of megarectum: pelvic US was carried out with full or partially full bladder with a convex multifrequency probe 3–5 Mhz. The US probe was positioned on the anterior abdominal wall in the midline, approximately 1–2 cm above the pubic symphysis at a 90° angle to the abdominal wall. This allowed the rectum to be visualized behind the urinary bladder as a crescent shape, which was measured in centimeters. Median rectal crescent size (transverse diameter) in children with rectal fecal retention was found to be ≥30 mm, whereas in healthy children it was not visualized or <30 mm [12]. In this study, we considered megarectum a median rectal transverse diameter over 40 mm. High resolution anorectal manometry with three-dimensional (3D) sphincter and anal reconstruction [13]: the study was performed with ManoScan^®^ Anorectal High-Resolution Manometry (3D HRAM 360/3D, Medtronic, Dublin, Ireland). Data acquisition, display, and analysis were performed with ManoView Software. With 3D HRAM, we evaluated:-Mean resting anal pressure;-Maximum voluntary contraction during squeeze maneuver (reported as increment vs. resting pressure);-Sphinterial asymmetry (difference of resting and squeeze pressure >20% between four cardinal anal segments evaluated with 3D analysis);-Rectal sensitivity (ml volume inflated in intrarectal balloon which determines a first defecatory urge);Volumetric enema: simulation of TAI under fluoroscopy in order to evaluate the volume of water (contrast) needed to opacify the entire left colon and the volume (number of puffs) of the rectal catheter needed to obtain the adhesion of the balloon to rectum walls without water leak during the procedure and without pain during and after insufflation.Training the caregiver, first using a simulator and subsequently on the patient [10].Beginning of the test treatment for 10 days, on the basis of carried out investigation we established: the volume of water to be infused in the colon and number of puffs needed to inflate the balloon catheter.

During this period, the TAI is performed every other day, noting in a diary format all parameters set—the position preferred by the patient during TAI (supine decubitus, lateral on the side, or directly on the toilet); the evacuation time; the need to induce evacuation with enema; and any problems such as pain, leakage of water during the procedure, sensation of complete evacuation, and episodes of fecal incontinence during the day or the night after or between the procedures.
g.Verification assessment after 10 days. At that time, after the evaluation of the diary format filled in, adjustments can be made as follows:-Modification of water volume (decrease in case of fecal incontinence following the procedure and increase in case of lack of benefit and/or incomplete evacuation);-Definition of the number of puffs required not to cause water losses during the procedure in absence to pain (pain assessed using validated pediatric scales);-Definition of mean time to evacuate after TAI;-Need for therapy with macrogol (if hard stools with painful defecation are passed);-Definition of timing of TAI (daily, every other day, or twice a week);h.Beginning the definitive TAI treatment.i.Clinical assessment at 1, 3, and 6 months with score.j.Rectal ultrasound after 1 month (T1) to check the absence of megarectum in case of efficacy of TAIk.Evaluation with anorectal manometry after 3 months in order to evaluate the improvement of anal resting pressure and dyssynergy. In case of persistence of manometric anomalies, the patient will undergo adjuvant treatment with biofeedback (BFB)—passive electrical stimulation/active contraction and relaxation cycle exercises depending on the anomaly (anal hypotonus or dyssynergy).

The assessed outcome included:-Relation among clinical, manometric, radiologic, and ultrasound parameters;-Correlation between the instrumental assessment and the outcome of patients considering different pathologies (need of adjustment after training period and rates of efficacy, complications/problems during treatment).

### 2.3. Statistical Analysis

Statistical analysis of quantitative and qualitative data, including descriptive statistics, was performed for all items. Continuous data were expressed as mean ± standard deviation (SD), unless otherwise specified. Frequency analysis was performed using the Pearson’s chi-square test and Fisher’s exact test as needed. The univariate analysis of variance (ANOVA) was performed for parametric variables to evaluate the differences between the different patient groups, and post hoc analysis with the Tukey test was used to determine whether there were pairwise intragroup differences. The repeated-measures ANOVA test was used to compare continuous variables at different time intervals. The correlation analysis was performed by Spearman rank correlation test. Data were analyzed by IBM SPSS Software 22 version (IBM Corp., Armonk, NY, USA). All *p*-values were two-sided and *p* < 0.05 was considered statistically significant.

## 3. Results

The demographic data and pre-treatment parameters with differences between pathologies are reported in Table 1.

In the ARM group, the patients had rectal bulbar fistula (four patients), rectal vaginal fistula (two patients), rectal bladder fistula (four patients), rectal prostatic fistula (two patients), and perineal fistula (two patients); six ARM patients had associated spinal anomalies partly responsible for the symptoms (two perineal fistula, one vaginal fistula, two rectal bulbar fistula, and one bladder fistula). In the HD group, two patients had long segment disease, whereas the other 10 patients had recto sigmoid disease (symptoms related to both anal hypertonus and slower peristalsis of the pull through segment or retentive attitude for pain from anal dermatitis or other); in all HD patients with obstructive symptoms after surgery, the presence of a residual aganglionic tract is first excluded. In the NI group, all patients had a severe neurogenic bowel associated with spastic and non-spastic tetrapharesis due to perinatal asphyxia. In the FFI group, all patients had a retentive or not retentive fecal incontinence according to Roma IV criteria [14].

No differences among different groups regarding T0 score were found: actually, all patients showed severe fecal incontinence with indication to bowel management with TAI.

In all groups, the rectal T0 diameter was pathological, with diagnosis of megarectum, with a mean diameter of 46.6 mm; the differences of rectal diameter were not statistically significant among groups, even though the highest rate was observed in the ARM group. 

As for manometric evaluation, sphincteric anomalies were observed more frequently in the ARM group, with lower values of anal resting pressure and asymmetry (Figure 1); the maximum squeeze pressure was more compromised in NI group, whereas the rectal sensitivity was lower in ARM group and greater in NI group. Rectal-sphincteric dyssynergy has been observed in 37% of all patients and was more evident in FFI group, even if the difference with other groups was not statistically significant. 

Regarding the enema parameters, the mean initial water volume needed to reach the left colon was initially 180 mL with lower volumes needed in ARM group Figure 2. The treatment outcome data are reported in Table 2.

After a 10-day test, treatment of 62.9% patients needed one or more adaptations of TAI parameters, mainly for the volume of water used for irrigation and the number of puffs required to inflate the rectal balloon. Among the ARM patients, 71% reported fecal/water incontinence after the procedure and the volume of water was reduced to an average of 120 (significantly lower if compared with other groups), whereas the FFI group required a higher mean volume of water if compared with all other groups; 85% of patients in ARM group performed the procedure by inflating the balloon with no or with 1 puff, whereas most of the patients in NI e FFI groups inflated with 2 or more puffs. This result significantly correlated with the volume causing urge defecation at manometry (*p* 0.000).

Macrogol was taken by all groups with no differences. After all adjustment, the 80% of all patients reported complete evacuation with a mean evacuation time of 26 min, without differences among groups. This data was confirmed by T1 ultrasound rectal evaluation, that showed the absence of megarectum in all groups, with a mean rectal diameter of 24 mm. The differences between the rectal diameter at T0 and T1 were statistically relevant both as a whole and within each group (ARM *p* 0.001, HD *p* 0.020, NI and FFI *p* 0.000).

Regarding clinical score, in each group, the improvement was statistically significant—as highlighted by T1, T3, and T6 scores (*p* 0.000); considering the whole treatment, until T3, the score values were statistically different among groups both for average and for trend (Figure 3), but this difference disappears after 6 months, when all groups showed high and similar values of score (*p* 0.171). At T1 and T3 evaluation, the ARM group showed lower values of score if compared with all other groups, whereas in the FFI group a faster improvement was observed.

The main factor influencing the response to treatment was the rate of sphincter anomalies: actually, patients with sphincter anomalies—regardless of the underlying pathology—had a worse trend, both for lower score values and for a slower improvement than patients without these anomalies at all time evaluation (T0, T1, T3, and T6 with *p* of 0.000, 0.001, 0.003, and 0.060 respectively) (Figure 4). The BFB positively influenced the trend of score at T6 time [*p* 0.003], with higher score values for patients who underwent this treatment. No differences were found among previously and new treated patients.

During all treatment, no complications or problems were observed and no patient abandoned the procedure.

## 4. Discussion

Transanal irrigation (TAI) is a non-surgical alternative which empties the bowel from approximately the rectal ampulla to the splenic flexure, thereby preventing fecal leakage between treatments. The idea is to keep the colon empty for suitable periods of time, in order to regain its propulsive ability and reestablish the control of defecation [1,2,3,4,15,16]. The efficacy and success of TAI with the use of Peristeen^®^ has been widely demonstrated also in children, with description of a standardized approach [10]. However, while some authors report constant results of efficacy in short- and long-term studies, in some cases a reduction of the success rate during follow-up is described [7,10]. In the literature, several studies show the high efficacy of Peristeen^®^ in several organic diseases, as well as in functional disorders [17,18,19,20,21,22,23]; however, to our knowledge, there are no studies comparing the effectiveness of TAI in patients affected by different pathologies. Furthermore, the actual best practices, which are very useful in the management of children with fecal incontinence, originate from protocols defined on adult patients. Instead, we think that this treatment should not be standardized always using same parameters, because each pediatric patient has different physio pathological peculiarities, which depend on the underlying pathology, on his/her medical history, and on personal and familiar compliance. The main reported problems are usually fecal/water incontinence during or after the procedure and pain during inflating of the rectal balloon; their presence, even they can be not considered real complications, can lead over time to the procedure presenting an impaired quality of life [7]. It is also mandatory to consider that popularity of transanal bowel management technique is influenced by an important cultural aspect: in different part of the world, families may not accept the use of transanal therapies which might be considered as violating the child and so, it is important improve the efficacy of this procedure as much as possible in order to propose it to families as a valid instrument to improve their quality of life. 

Furthermore, no clinical parameters seem to predict the response to treatment and, to date, no dedicated instrumental evaluations are described for patients undergoing TAI.

The aim of our analysis was to demonstrate the importance of an accurate work up—both clinical and instrumental—of patients to submit to Peristeen^®^, in order to achieve a tailored approach for each patient, which is the key to increasing efficacy and adherence to the treatment.

We included 24 patients previously treated with TAI without benefit (patients using enema or anal plug), calling them back to retry the procedure framing it in the new evaluation protocol: the efficacy obtained among these patients is the proof that this protocol is really effective compared to using TAI without an instrumental management first.

Our results clearly show that the evaluation of all patients in pre-treatment time with rectal ultrasound, volumetric enema, and anorectal manometry was effective; actually, this allowed us to establish the correct volume of water to be used in order to avoid incontinence and the number of puffs to be used to insufflate the balloon in order to avoid pain, discomfort, or leaking during the procedure. Besides, anorectal manometry with 3D reconstruction allowed us to identify sphincterial anomalies, which were found to be the most important prognostic factor for TAI efficacy. Our results showed that parameters differ among the various groups and, within each group, among patients.

In long-term follow-up of ARMs, fecal incontinence (FI) and severe constipation remain the most frequent and disabling postoperative clinical problems because of their impact on the quality of life. The postsurgical management of these anorectal deficits can be very difficult, and TAI is therefore often indicated in order to improve outcomes [24,25,26]. Our results showed that patients with ARM need overall lower water volume and lower number of puffs (from 0 to 1), probably due to shorter colon, altered rectal sensitivity, intrinsic motility disorders, and sphincterial anomalies; almost all patients (85.7%) needed adjustment of parameters; while achieving good results at 6 months, their improvement is slower over time than in other groups. If standard protocol described in the literature were used, as well as done by ourselves in previously treated patients, these fragile patients would probably suffer discomfort, resulting in a refusal of the procedure. It is therefore mandatory to describe to the patients and their families the effectiveness of the procedure, and the need to personalize it, indicating the prognosis and any negative effect. 

The most commonly reported long-term functional problems after definitive surgical management of Hirschsprung’s disease are constipation, fecal incontinence, and enterocolitis. TAI is well described in the comprehensive management of these patients [27,28]. Our results showed that HD patients had an intermediate trend, better than ARMs patients but worse than NI e FFI patients. These patients can show sphincterial anomalies often due to iatrogenic causes, but their squeeze contraction pressure is overall good and they can therefore better retain stool and water during TAI, better tolerate the inflated balloon because their rectal sensibility is overall normal; they tend to keep a larger rectal diameter at T1 evaluation, probably because they mainly referred to obstructive symptoms with severe constipation; for patients with dyssynergia, the BFB treatment improves the long-term results even more.

Modulation of the neural activity in the enteric nervous system by extrinsic innervation is abnormal in children with cerebral disorders (NI group); thus, it is not surprising that insults to the brain may result in significant dysfunction of gastrointestinal tract, which primarily shows as a gut dysmotility. Chronic constipation is a common problem in children with disabilities, with estimated prevalence varying from 26% to more than 50% of children with severe disabilities and TAI can be indicated as a second line of treatment after macrogol or standard enema [29,30]. Our data show that these patients get a complete evacuation, emptying of the whole left colon with TAI for long time, using high water volume, tolerating the balloon inflated with 2–3 puffs, and greater outcomes are observed if macrogol is associated; their anal tone can be low due to persistent megarectum or to neurological anomalies; overall, they have a good clinical response, improving in all cases at 3- and 6-month evaluation with a substantial quality of life improvement. 

Functional fecal incontinence (FFI) has a significant negative impact on the child’s quality of life, regarding both physical and psychological aspects. Approximately 50% of all children with FC respond to standard treatment within the first 6–12 months, but a significant percentage of non-responders may require a more invasive approach using TAI [14,31,32]. Our results show that these patients have a fast and positive response to TAI, with a satisfactory score already after 1 month and better results after 6 months if compared to all other groups, probably because the sphincterial anomalies are not often observed and have a functional cause secondary to rectal stasis. They often show a very dilated and tortuous colon and require a larger water volume to irrigate colon until left flexure; they need macrogol and BFB in 60% and 50% of cases respectively, with complete evacuation and success in 90% of cases with a shorter evacuation time than other groups. Considering the elevated efficacy and the quick results, TAI can be proposed as a valid bowel management treatment also in patients with FFI. 

Furthermore, our results showed that patients with dyssynergic defecation receiving biofeedback would exhibit greater improvement in bowel symptoms and anorectal physiology than those receiving only TAI. For children with fecal incontinence resulting from sphincter dysfunction, biofeedback training to strengthen the peripheral anal muscle appears to be efficacious and this result improves the quality of life even further. It can be considered appropriate when specific pathophysiological mechanisms are known and the voluntary control of responses can be learned with the aid of systematic information about functions not usually monitored at a conscious level [33,34,35,36].

Considering all the results described in our study, even though the small number of patients is insufficient to draw firm conclusions, we are strongly convinced of the need for a tailored TAI procedure, using an advanced protocol based on clinical score evaluation, diary, anorectal manometry, and volumetric enema. No patient abandoned the procedure, the problems were prevented, and the efficacy was very good, with a complete continence and cleaning after 3–6 months of treatment. 

Our future goal is to confirm the validity of our protocol on a larger number of patients, reducing the problems related to TAI as much as possible, increasing its effectiveness, and confirming the differences highlighted so far in the different pathologies.

## 5. Conclusions

TAI can be proposed as a valid bowel management treatment for different pathologies, but without standardization and with different parameters tailored for each patient, we should remain aware that the responses will be different among patients and over time. In order to achieve greater efficacy, it is important to evaluate the patients in pretreatment setting, tailoring the TAI mainly with volumetric enema and anorectal manometry.

The aim of our study was demonstrate the importance of a tailored approach and we demonstrated this precisely because we have compared the individual pathologies with each other: every pathology and every single patient needs a personalized treatment and we strongly believe that it is not correct to approach all patients in the same way. Only after having understood this with a preliminary study, it is possible to subsequently start studies on a single pathology, including more patients.

## Figures and Tables

**Figure 1 children-08-01174-f001:**
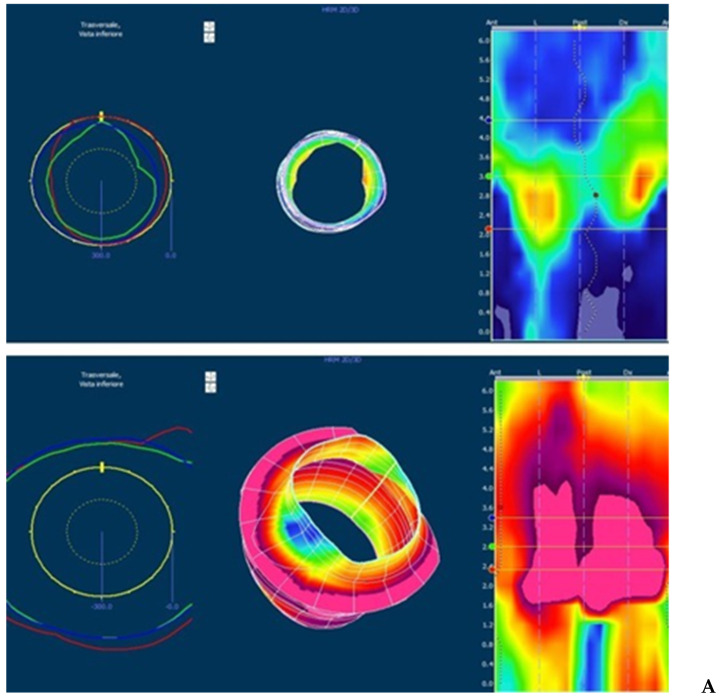
3D high-resolution anorectal manometry (3d HRAM) in patients without (**A**) and with (**B**) sphincter anomalies. The image shows the 3D reconstruction of the anal canal at rest and after squeeze (images below). (**A**) Normal sphincter with good pressure values and symmetry; the pressure increment after squeeze is good for amplitude and symmetry. (**B**) Severe anal hypotonia and asymmetry with lower pressure in anterior, right and left segments at rest; during squeeze, the pressure is higher but not present in all segments.

**Figure 2 children-08-01174-f002:**
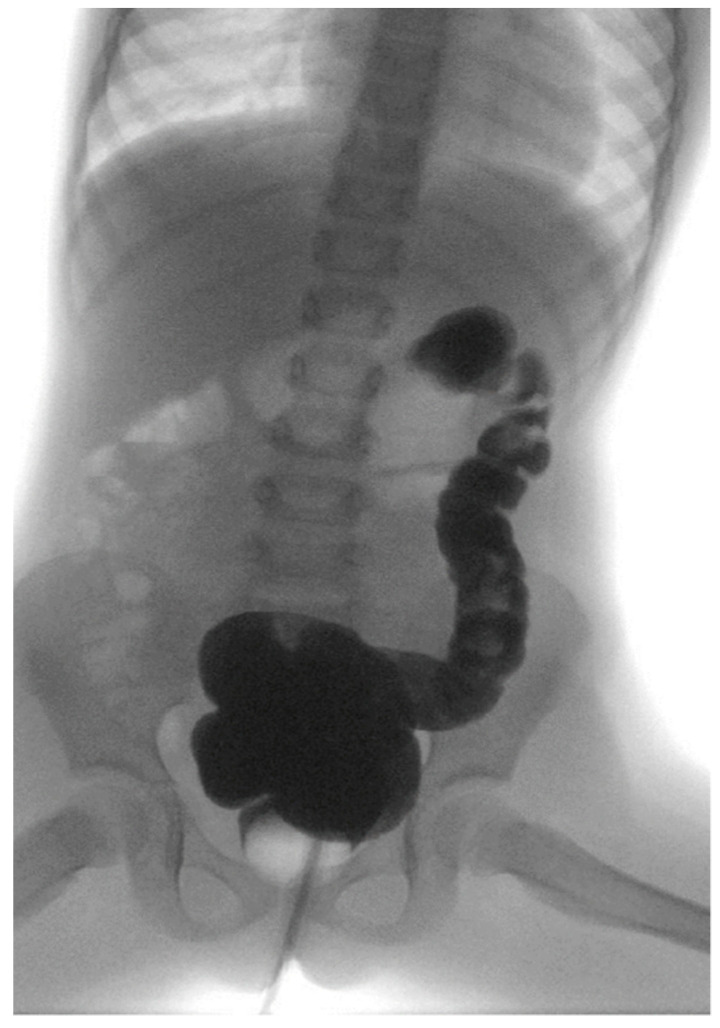
Volumetric enema with contrast until left colon.

**Figure 3 children-08-01174-f003:**
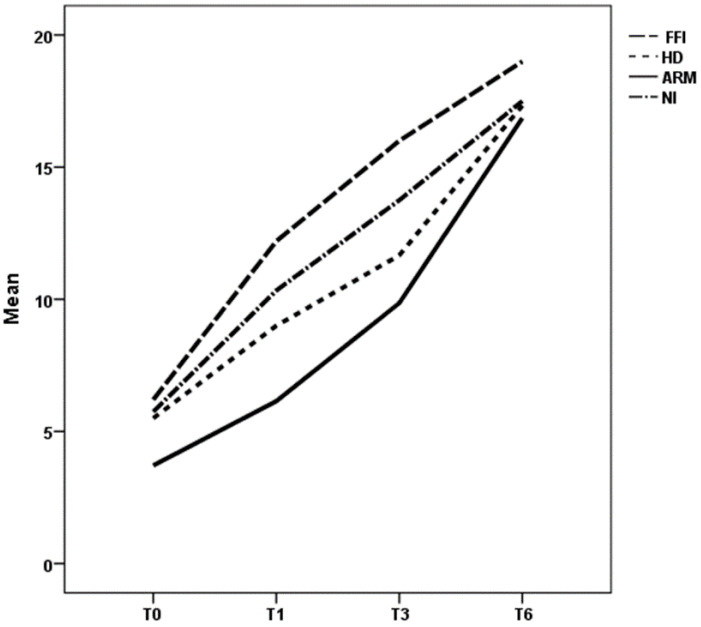
Trend of mean clinical scores at T0, T1, T3, and T6 for different groups. ARM: anorectal malformation; HD: Hirschsprung’s disease; NI: neurological impairment; FFI: functional fecal incontinence.

**Figure 4 children-08-01174-f004:**
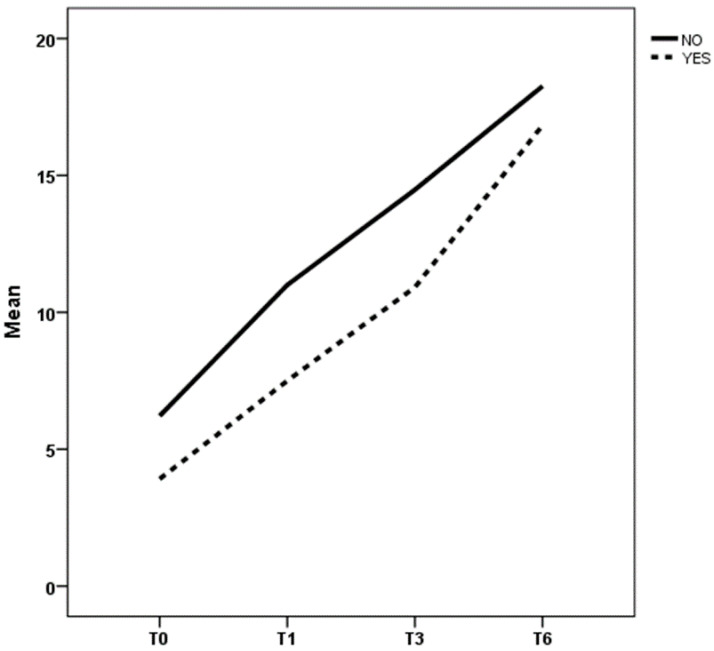
Caption. Trend of mean clinical scores at T0, T1, T3, and T6 considering overall sphincteric anomalies in all patients. YES: patients with anomalies; NO: patients without anomalies.

**Table 1 children-08-01174-t001:** Pretreatment parameters in each group.

	ARM	HD	NI	FFI	TOT (Range)	*p*
No. of patients (%)	14 (20)	12 (17.1)	24 (34.3)	20 (28.6)	70	
Age (years) (mean ± SD)	10.29 ± 3.25	6.67 ± 3.20	10.58 ± 3.96	8.30 ± 1.7	9.20 ± 3.394 (r 4–17)	0.006 * 0.021 § 0.005
Megarectum (No. patients and %)	10 (71%)	8 (66%)	8 (33%)	12 (60%)	38 (54.3%)	0.380
Rectum diameter (mean ± SD)	46.29 ± 11.6	47.50 ± 12.14	44.58 ± 15.1	49.0 ± 16.6	46.69 ± 14.01 (r 25–80)	0.912
**Rectal manometry**						
Sphinteric anomalies (No. patients and %)	12 (85.7%)	6 (50%)	4 (16.6%)	2 (10%)	24 (34.3%)	0.004
ARS (mmHg) (mean ± SD)	36.14 ± 12.94	60.67 ± 32.60	74.17 ± 19.8	78.50 ± 18.7	65.49 ± 25.77 (r 18–110)	0.001 * 0.004 § 0.002
MSP (mmHg) (mean ± SD)	34.29 ± 8.3	76.67 ± 22.28	25.83 ± 10.83	58.00 ± 20.97	45.43 ± 24.89 (r 10–110)	<0.0005 § 0.035 ^°$ <0.0005
Urge (ml vol) (mean ± SD)	58.57 ± 16.7	88.33 ± 28.5	106.67 ± 21.46	96.00 ± 27.56	90.86 ± 28.83 (r 40–140)	0.002 § 0.020 * 0.001
Dissinergy (No. patients and %)	4 (28.5%)	4 (33.3%)	6 (25%)	12 (50%)	26 (37.1%)	0.387
**Enema parameters**						
Water volume (mL) (mean ± SD)	164.29 ± 37.7	216.67 ± 25.8	200 ± 60.3	235 ± 33.7	180 ± 48.80 (r 100–250)	0.024 § 0.019

ARM: anorectal malformation; HD: Hirschsprung’s disease; NI: neurological impairment; FFI: functional fecal incontinence constipation; SD: standard deviation; ARS: anal resting pressure; MSP: maximum squeeze pressure; vol: volume. TOT: total Statistical analysis. The mean difference is significant at the 0.05 level. Post hoc analysis * ARM vs. NI; § ARM vs. FFI; ^ ARM vs. HD; ° HD vs. NI; $ NI vs. FFI.

**Table 2 children-08-01174-t002:** Treatment outcome for each group.

	ARM	HD	NI	FFI	TOT (Range)	*p*
Adjustment after 10 days (no. patients and %)	12 (85.75)	8 (66.6%)	12 (50%)	12 (60%)	44 (62.9%)	0.477
Fecal incontinence (no. patients and %)	10 (71.4%)	8 (66.6%)	8 (33.3%)	10 (50%)	36 (51.4%)	0.366
Mean evacuation time (min ± SD)	28.57 ± 10.2	25.00 ± 11.8	30.00 ± 11.2	21.00 ± 11.0	26.29 ± 11.2 (r 10–50)	0.283
Definitive volume water (ml ± SD)	121.43 ± 26.7	175.00 ± 27.3	195.83 ± 49.8	205.00 ± 36.8	180 ± 48.8	0.001 * 0.002 § 0.001
Patients 1 puff	12 (85.7%)	8 (66.6%)	4 (16.6%)	8(40%)	32 (45.7%)	0.020
Patients 2 puff	2 (14.3%)	4 (33.4%)	20 (83.4%)	12 (60%)	38 (54.3%)	0.020
Complete evacuation (no. patients and %)	12 (85.7%)	10 (83.3%)	16 (66.6%)	18 (90%)	56 (80%)	0.711
Clinical Score T1 (mean ± SD)	6.14 ± 1.34	9.0 ± 3.34	10.3 ± 2.38	12.2 ± 1.9	9.8 ± 3.08	<0.0005 § <0.0005 * 0.004
Clinical Score T3 (mean ± SD)	9.8 ± 1.57	11.6 ± 3.88	13.7 ± 3.36	16.0 ± 2.1	13.2 ± 3.51	0.001 § 0.001 * 0.045 # 0.038
Clinical Score T6 (mean ± SD)	16.8 ± 2.2	17.3 ± 2.42	17.5 ± 2.2	19.0 ± 1.4	17.7 ± 2.1	0.171

ARM: anorectal malformation; HD: Hirschsprung’s disease; NI: neurological impairment; FFI: functional fecal incontinence; SD: standard deviation. Statistical analysis. The mean difference is significant at the 0.05 level. Post hoc analysis: * ARM vs. NI; § ARM vs. FUNC; # HD vs. FUNC.

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
