# Peer review of "Advanced Management Protocol of Transanal Irrigation in Order to Improve the Outcome of Pediatric Patients with Fecal Incontinence"

_children, 2021, doi:10.3390/children8121174_

Round 1
Reviewer 1 Report
The authors evaluated patients with fecal incontinence caused by different diseases. Functional tests and anatomy of the colon, rectum and anal canal were performed for initial evaluation and during follow-up of these patients while treated with a systematized Transanal Irrigation protocol. Fecal incontinence is a great burden for patients and their families, and until today, the available therapeutic methods have shown unsatisfactory results. TAI has been widely used to improve the quality of life of these patients, but its use is not supported by functional and anatomical studies that can define individualized bases for the procedure. The authors used the best methodological tools for anatomical and functional evaluation of the colon and segments of patients with fecal incontinence. The evaluation is original. Morpho-functional studies in children are very scarce.
Author Response
TO REVIEWER# 1: thank you for the positive comments on our report. We really appreciated your comments. We agree with you because, although TAI has been widely used to improve fecal incontinence and quality of life of children and their family, morpho functional studies in children are very poor; we think that our protocol, confirming it in the future with a large number of patients, can improve further the efficacy of TAI.
Reviewer 2 Report
Thank you for submitting this paper on a relevant subject, such as the use of Trans-anal irrigation in the management of Colorectal and pelvic disorders, with the aim of creating a protocol of use, tailored on specific patients and their needs.
Few comments: (it should be acknwoledged that the popularity of trans-anal bowel management technique is influenced by an important cultural aspect. In different parts of the world, families may not accept the use of trans-anal therapies as considered as violating the child).
1) Introduction section:
The author should avoid contractions in the manuscript, unless previously used and specified (Introduction paragraph, line 33: pts)
The author should clarify the statement re use of TAI when failure of conservative managament: TAI is still considered as a conservative management of fecal incontinence and constipation, and it is an estabilished and recognized part of bowel management strategies in the treatement of colorectal and pelvic disease (line 35: TAI as an alternative to conservative medical therapy)
The author should cite with references the line 38: In recent years, some data have been published concerning the efficacy of trans-anal irrigation in the pediatric population.
2) Material and Methods section:
Line 61, the Author should clarify and better explain the meaning of problems that lead to the failure of use of TAI, such as: non compliance of patient/family, pain, persisting incontinence. In our experience, the pain and the non compliance with trans-anal approaches is the most common cause of failure. The author should clarify if in their department, there is the option of access to different devices- such as Qufora- system that doesn't have a balloon device but it's a conic designed device which solves the issue with balloon inflation related pain. The author should mention which alternative strategies for the 24 patients -who failed the TAI regime- have been used (?ACE)
3) Discussion:
Line 233: Reduction of success rate in the use of peristeen is described- needs reference
Line 262: the author may want to consider that TAI is indicated to improve patient's outcome rather than bowel outcome
Line 270:the author may want to consider to change with: patients and their families
Children with neurological disabilities: this specific class of patients can become quite challenging especially when it comes to technicalities such as: families to deal with mobilization of these children, very so often on wheelchairs. While the use of Antegrade colonic enemas, sound more feasible in this specific pool of patients- unless Mitrofanoff in situ- can the authors describe in more details their approach to this specific class of patients?
Would the authors consider to create a flow chart to summarize the advised/proposed approach algorithm- which would enrich the conclusion?
Author Response
TO REVIEWER# 2:. Thank you for your comments; we appreciated and followed your suggestions and modified our paper in accordance with them, as specified in the point-by-point reply. We have highlighted the changes made in yellow in the text
- We added in the text (discussion section) that “popularity of trans anal bowel management technique is influenced by an important cultural aspect: in different part of the world, families may not accept the use of trans anal therapies as considered as violating the child and so, it is important improve as much as possible, the efficacy of this procedure in order to propose it to families as a valid instrument to improve the quality of life.
- Introduction section: we added “pts” after first use of patients
- We agree with you that TAI is to be considered as a conservative management but its use is usually not proposed immediately but only after failure of more conventional treatments such as enema and laxatives. We clarify this in the introduction text
- We added the reference (line 38)
- In the Material and Methods section, we clarify the problems that lead to be failure mainly pain during procedure, incontinence after procedure and incomplete evacuation). We have no experience with different devices. Only 1 patient was submitted to ACE while the remaing patients continued treatment with “standard” enema or only anal plug; these patients were still in follow up at our clinic and were called back to retry TAI procedure framing it in the new evaluation protocol. The efficacy obteined in these our pts is the proof the the protocol is really effective compared to using TAI without an instrumental management first. We added this comment in the discussion.
- In the discussion we added the reference (line 233)
- The sentence on line 262 was modified
- The sentence on line 270 was corrected
- We agree with you that children with neurological disabilities represents a challenge as long as they are very problematic patients often already subjected to treatments such as gastrostomy or tracheostomy that parents would like to hospitalize as little as possible for fear of long hospital stays without the possibility of real changes to their quality of life; in our center these patients are evaluated in a dedicated multidisciplinary clinic: regarding the bowel aspect, they are pts who use high dose laxatives often without benefit and the parents stimulate and empty the stool manually. Especially by the human point of view these families report great benefit in the use of TAI because with a simple procedure they improve a very important aspect for the quality of life of their children. In the manuscript we don’t describe this “human” aspect to avoid stretching the text too much ma we think very much for this comment.
Reviewer 3 Report
The manuscript described the management protocol of transanal irrigation for the children with fecal incontinence. Unfortunately, this manuscript does not seem to have enough quality to be published from the journal for the reasons described below.
・The structure of this manuscript is not appropriate (e.g., “4 Discussion … 4.1. Discussion…). This manuscript is very difficult to read and evaluate.
・Explanation of the table 2 which show the main results of this manuscript could not be found in the manuscript.
Author Response
TO REVIEWER# 3:. Thank you for your comments; we are very sorry for your negative opinion. We hope that the changes made to the structure will be useful for your evaluation.
- Unfortunately, the structure of the manuscript has been altered by errors in the numbering of paragraphs. We corrected these errors and revised the points of each paragraph
Tables and figures have been moved to the end of the results in order to make the reading of the paragraph smoother
Reviewer 4 Report
This is an interesting article; however, I would like to make a few comments and suggestions.
1. The summary must be independent, so I think it would be good to specify what each abbreviation means.
2. Table1. The mean age for the HD group was 6.67 ± 3.20 years. It is hard to believe that a 3 or 4 year old child cooperates and answers correctly to such requirements / questionnaire (the need to induce evacuation with enema, any problem as pain, sensation of complete evacuation).
3. Line 148-150. In the ARM group, it is unusual that a child with perineal fistula or even vaginal fistula to suffer any continence problem or constipation. Did those children have other associated problems?
4. Line 150-151. Also for HD group, usually the recto-sigmoid form respond well after surgery and is unusual for the patients to present with fecal incontinence and/or bowel disfunction unresponsive to conventional treatments with laxative and enema.
5. Line 154 - for Roma IV criteria, a short description or at least a reference is needed.
6. I do not think it is correct to compare the groups of ARM and HD patients with the other two groups, because the first two were treated surgically at the perineal level, which probably affected the sphincter innervation. The mechanisms of occurrence of defecation problems are therefore different, and the treatment must be personalized.
7. I do not see the point of introducing a subchapter of Discussions (4.1) inside chapter 4 - Discussions.
8. I suggest a more careful selection of patients and reinterpretation of the results. I think that the comparison between groups of patients with different mechanisms of defecation problems can lead to a misinterpretation of the results.
Author Response
TO REVIEWER# 4:. Thank you for your comments, we appreciated comments and we modified according your suggestion.
- The summery with list of abbreviation was added at the end of the manuscript (supplementary materials)
- For children under 5 yrs (certainly not very cooperative) we considered the answers given by parents.
- We better specified the association between ARM and spinal anomalies as responsible to symptoms even with not severe malformations
- For HD pts faecal incontinence is mainly caused by severe retention; theese pts often have a severe sphincter anal hypertonus, anomalies of peristalsis that cause slow transit in the pull through segment or retentive attitude for anal pain, dermatitis or other. We added this comment in the text
- The references for Roma IV criteria is n° 30 and we added this reference in the text
- We agree with you that it not correct compare pts with and without perineal surgery but our number are too small in order to have significant results; our evaluation highlighted how pts with sphincter anomalies have a different outcome than those without alterations: knowing this it is possible to give a more real prognosis of response to treatment.
- All points of paragraphs are revised; we became aware of an error in the numbering of the paragraphs and subchapter; tables and figures have been moved to the end of the results in order to make the reading of the paragraph smoother
- In this preliminary study we wanted to compare all patients affected by pathologies that usually require TAI in order to confirm that each of them benefits from a tailored treatment precisely because they have peculiarities given by its underlying pathology.
Round 2
Reviewer 3 Report
Thank you for re editing and rewording.
Author Response
Thanks for your approval; we are pleased that the comments and changes have been effective for you
Reviewer 4 Report
The topic of the article is interesting, but I think there is a bias in patient
selection. I think that patients under 5 years of age should not be introduced in the study because they do not cooperate; I do not want to say that TAI would not be a very good method, only that very young patients cannot give correct answers. Also, I think that there are patients from HD group
to whom TAI treatment is not absolutely necessary; first try enemas and laxatives; if not work, then surgery was not well performed; maybe the agangliotic segment was not completely removed. Then, the comparison between groups of patients with different mechanisms of defecation problems can lead to a misinterpretation of the results.
I suggest rewriting the manuscript with a more careful selection of patients and reinterpretation of the results.
Author Response
TO REVIEWER# 4: We appreciated your comments on certainly very relevant aspects regarding our protocol for TAI procedure. We therefore want to clarify some aspects better:
- The use of TAI, as approved by NICE committee, is foreseen for children under 5 yrs; we excluded from this study and in general we never submit to TAI, uncooperative children. This is one of the reasons why there are few patients.
- In order to evaluate discomfort during procedure we use dedicated pediatric scale of pain and parents give answers on objective parameters as incontinence during or after procedure, discomfort during procedure, complete/abundant evacuation: all data that parent who carried out the procedure certainly knows how to recognize, describe and report.
- About HD patients, in our centre, for all patients with obstructive symptoms we always first exclude the persistence of an aganglionic tract; the first line of treatments are laxative and enemas: when patients benefit form enemas but are addicted to them or unresponsive, we refer to TAI.
- We have included too few patients to be able to assess the benefits of TAI for each disease in a meaningful way. Aim of our study was demonstrate the importance of a tailored approach and we demonstrated this precisely because we have compared the individual pathologies with each other: every pathology and every single patient needs a personalized treatment and we strongly believe that it is no correct to approach all patients in the same way. Only after having understood this with a preliminary study, it is possible to subsequently start studies on a single pathologies , including a large number of patients (however , studies already present in the literature)
Round 3
Reviewer 4 Report
Dear authors, the explanations you gave me at the last evaluation are relevant and I appreciate them. Please include the first three explanations given by you in the Results chapter, and the fourth in the Discussions chapter. This would greatly improve the understanding of patient selection, the results obtained and, implicitly, the quality of the article.
- "The use of TAI, as approved by NICE committee, is foreseen for children under 5 yrs; we excluded from this study and in general we never submit to TAI, uncooperative children. This is one of the reasons why there are few patients.
- In order to evaluate discomfort during procedure we use dedicated pediatric scale of pain and parents give answers on objective parameters as incontinence during or after procedure, discomfort during procedure, complete/abundant evacuation: all data that parent who carried out the procedure certainly knows how to recognize, describe and report.
- About HD patients, in our centre, for all patients with obstructive symptoms we always first exclude the persistence of an aganglionic tract; the first line of treatments are laxative and enemas: when patients benefit form enemas but are addicted to them or unresponsive, we refer to TAI.
- We have included too few patients to be able to assess the benefits of TAI for each disease in a meaningful way. Aim of our study was demonstrate the importance of a tailored approach and we demonstrated this precisely because we have compared the individual pathologies with each other: every pathology and every single patient needs a personalized treatment and we strongly believe that it is no correct to approach all patients in the same way. Only after having understood this with a preliminary study, it is possible to subsequently start studies on a single pathologies , including a large number of patients."
Author Response
We appreciated your final approval and we modified manuscript by adding further details as you suggested (additions highlighted in yellow).
. We hope that these additional clarifications provided will be sufficient and that the manuscript will now suitable for publication in the journal.